# BiMix: Bivariate Data Mixing Law for Language Model Pretraining

## Abstract

Large language models have demonstrated remarkable capabilities across various tasks, primarily attributed to the utilization of diversely sourced data. However, the impact of pretraining data composition on model performance remains poorly understood. This paper introduces **BiMix**, a novel bivariate data mixing law that models the joint scaling behavior of domain proportions and data volume in LLM pretraining. **BiMix** provides a systematic framework for understanding and optimizing data mixtures across diverse domains. Through extensive experiments on two large-scale datasets, we demonstrate **BiMix**'s high accuracy in loss extrapolation (mean relative error <0.2%) and its generalization to unseen mixtures ($R^2$ >0.97). Optimization of domain proportions yields superior model performance compared to existing methods. Furthermore, we establish entropy-based measures as efficient proxies for data mixing, offering a computationally lightweight strategy. Our work contributes both insights into data mixing dynamics and practical tools for enhancing LLM training efficiency, paving the way for more effective scaling strategies in language model development.

## 1 Introduction

Large language models (LLMs) have achieved remarkable success, revolutionizing capabilities for comprehending and generating human-like text across diverse applications, from question answering to code generation (Bubeck et al., 2023; OpenAI, 2024). As these models scale up, the composition of pretraining data becomes increasingly crucial for performance and generalization (Longpre et al., 2023). The emergence of multi-source datasets has presented both opportunities and challenges in LLM development (Gao et al., 2020; Shen et al., 2023; Chen et al., 2024), necessitating a deeper understanding of data mixing strategies.

Current approaches to data mixing often rely on heuristics (Touvron et al., 2023; Shen et al., 2023) or computationally expensive optimization techniques (Du et al., 2022; Xie et al., 2023; Fan et al., 2023). While these methods have shown promise, they require significant computational resources and lack a general framework for understanding the scaling behavior of mixed-domain training. The absence of a systematic approach to data mixing hinders efficient resource allocation and limits the ability to predict model performance across varied data compositions. Recent efforts have explored related techniques (Xia et al., 2024; Albalak et al., 2023; Shen et al., 2023), yet comprehensive and efficient solutions remain elusive.

The fundamental challenge lies in the complex interplay between different data domains in multi-source datasets. Existing research (Kaplan et al., 2020; Hoffmann et al., 2022) primarily focuses on scaling laws for individual metrics, overlooking this crucial aspect. This oversight hampers the progress towards more versatile models capable of excelling across multiple domains (Dong et al., 2024). A systematic mixing law remains to be developed to efficiently assess the importance of diverse data sources and understand their impact on model performance.

To address these challenges and fill the gap in current research, we introduce **BiMix**, a novel bivariate data mixing law that provides a systematic framework for understanding and optimizing data mixtures in LLM pretraining. Our approach is rooted in the observation that the scaling behavior of LLMs can be disentangled into two key components: domain mixing proportions and training data quantity (embodied by model training steps). By mathematically formulating the relationship

between these components and model performance, **BIMIX** offers a powerful tool for predicting and optimizing training outcomes.

We validate the proposed mixing law on two large-scale datasets, demonstrating its applicability across various scaling scenarios. Our experiments show that **BIMIX** not only provides accurate predictions of model performance across different data mixtures but also enables optimization of domain proportions, outperforming existing high-cost methods in terms of convergence speed and downstream task performance.

The key contributions are summarized as follows:

- A mathematically formulated mixing law that jointly models the scaling behavior of domain proportions and total training volume, providing good interpretability and functional extensibility.
- Comprehensive experiments demonstrating the effectiveness of **BIMIX** in predicting and optimizing model performance across diverse datasets and training scenarios.
- Empirical evidence supporting the use of entropy-based measures as lightweight mixing proxies, offering new insights into efficient data mixture optimization.

## 2 RELATED WORK

**Pretraining Data Mixtures** The coverage and diversity of pretraining data play significant roles in shaping the generalization capabilities of language models (Radford et al., 2019; Brown et al., 2020; Touvron et al., 2023). Data mixtures from multiple sources, such as the Pile (Gao et al., 2020) and ROOTS (Laurençon et al., 2022), are typically curated based on manually devised rules. However, the heuristics lack universal standards and portability. The GLaM dataset (Du et al., 2022) determined domain weights based on the component performance in a small model; however, specific details are not disclosed. SlimPajama-DC (Shen et al., 2023) investigated the effects of data mixtures using a set of predefined configurations and delivered several insights. Recently, DoReMi (Xie et al., 2023) and DoGE (Fan et al., 2023) proposed learning-based methods to optimize domain proportions by iterating between training reference and proxy models. These methods provide viable pathways but require considerable computational costs. In contrast, our study demonstrates that entropy proxies can produce data mixtures of comparable or even superior quality, while providing a more practical training-free solution. Besides, Chen et al. (2023) explored the effects of data sequencing from a curriculum learning perspective, whereas our research focuses on the concurrent integration of diverse data domains.

**Neural Scaling Laws** Investigations into the scaling behavior of neural models have spanned across domains such as computer vision (Klug & Heckel, 2023; Zhai et al., 2022; Jain et al., 2023; Sorscher et al., 2022) and natural language processing (Ivgi et al., 2022; Gordon et al., 2021; Ghorbani et al., 2022; Bansal et al., 2022). Kaplan et al. (2020) thoroughly evaluated the scalability of Transformer architectures across a wide range of model sizes and data volumes. Chinchilla (Hoffmann et al., 2022) identified similar scaling patterns through rigorous experimentation and suggested a slightly different configuration for compute-optimal pretraining. The impactful GPT-4 model (OpenAI, 2024) validated the predictive accuracy of scaling laws and underscored their important role in the development of large language models. Concurrently, additional research efforts seek to elucidate the principles governing scaling laws (Sharma & Kaplan, 2022; Michaud et al., 2023) and to investigate scaling effects on downstream tasks (Tay et al., 2022; Isik et al., 2024; Caballero et al., 2023; Cherti et al., 2023). In the context of data mixtures, Ye et al. (2024) proposed a composite exponential law to capture the interactions among domains; yet, its scalability is challenged by increased complexity for expanding domain numbers, as compared in Appendix C. Our study is distinguished by two key aspects: First, we introduce a scalable mixing law that accurately captures the scaling behavior associated with the composition of training datasets, demonstrating the modeling ability to up to 22 domains. Second, the proposed bivariate mixing law jointly models two input variables, *domain proportion* and *data volume*, thereby offering broader applicability.

## 3 THE PROPOSED **BIMIX**

Existing scaling law research primarily investigates the variation of a single scalar metric related to trained models concerning certain scaling factors. A prominent example is the relationship between

a model's validation loss and the amount of parameters or training tokens. However, in practice, the training datasets for large language models encompass diverse data domains, and these scaling laws only capture the predictability of the *averaged* validation loss across multiple domains. Since each domain corresponds to vastly different corpora, knowledge, and formats, modeling solely the average provides a coarse estimate of the model's performance and fails to reflect the predictability of individual domains.

In the context of data-centric language modeling, this study examines the scaling behavior of pre-trained models across finer-grained data domains. Notably, simply applying existing scaling laws to each domain is inappropriate, as the amount of training data for one single domain is not an independent variable; rather, it is determined by the total amount of training data and the proportion allocated to that domain, calculated as $|\mathcal{D}_i| = |\mathcal{D}| \times r_i$. The training data for a specific domain can be adjusted either by changing the total training data or by modifying the allocated proportion. Consequently, the data mixing modeling inherently involves a bivariate joint effect. Moreover, when the total amount of training data is fixed, any change in the proportion allocated to one domain will also affect the proportions (and thus the training amounts) of all other domains. This interdependence among domains highlights the need for dedicated research on the scaling laws of data mixing.

## 3.1 FORMULATION

Assume the training dataset consists of $m$ domains, each allocated a proportion $r_i$ under the unit-sum constraint:

$$\vec{r} = (r_1, r_2, \ldots, r_m) \quad \text{subject to} \sum_{i=1}^{m} r_i = 1. \tag{1}$$

We propose **BIMIX** as a system of equations to model the vectorized scaling laws of data mixing across the $m$ related domains in terms of the domain proportions $\vec{r}$ and training steps $s$:

$$\vec{L}(\vec{r}, s) = (L_1(r_1, s), L_2(r_2, s), \ldots, L_m(r_m, s)). \tag{2}$$

In this context, the validation loss of the model on the $i$-th domain, given the domain's proportion $r_i$ and the number of training steps $s$, is defined as follows:

$$L_i(r_i, s) = \frac{A_i}{r_i^{\alpha_i}} \left( \frac{B_i}{s^{\beta_i}} + C_i \right) \quad \text{for } i = 1, 2, \ldots, m, \tag{3}$$

where the constants $A_i$, $B_i$, $C_i$ and the exponents $\alpha_i$, $\beta_i$ are coefficients to be fitted. Notably, only five fitting coefficients per domain are needed to capture the joint scaling behavior concerning both the mixing proportion and the total training volume. This linear scalability with the number of domains offers a significant advantage over the quadratic complexity of other modeling approaches, substantially reducing the number of observational data points required for fitting. A detailed discussion of the complexity analysis can be found in Appendix C.

## 3.2 OBSERVING SCALING BEHAVIORS BY DISENTANGLING VARIABLES

It is important to recognize that scaling laws are fundamentally empirical formulas, providing mathematical descriptions that closely approximate real-world scaling phenomena. The construction of our proposed Eq. (3) is informed by a disentangled observation of the scaling behaviors of two variables, designed to offer strong interpretability. Next, we will elaborate on the observed scaling behaviors along with intuitive visualizations from the perspectives of the two input variables, as well as the considerations that guided the derivation of the final functional form.

**Scaling Training Steps Under Fixed Domain Proportions**    Figure 1 showcases the scaling behavior of the input variable $s$ on the Pile and SlimPajama datasets. Both the x and y axes are visualized on a logarithmic scale. Each line in the subplots corresponds to a specific value of the current domain proportion $r_i$, and the six lines represent the outcomes of six model training sessions on different data mixtures. These lines illustrate how the validation loss of each domain changes as the training steps increase. The discrete points in the figure indicate the actual evaluation results of the trained model, collected every 5 billion training tokens, while the dotted lines represent the fitted Eq. (3) derived from these observational data points. Overall, it is clear that the proposed mixing law fits the observed data points closely, and the curves exhibit a consistent pattern of downward

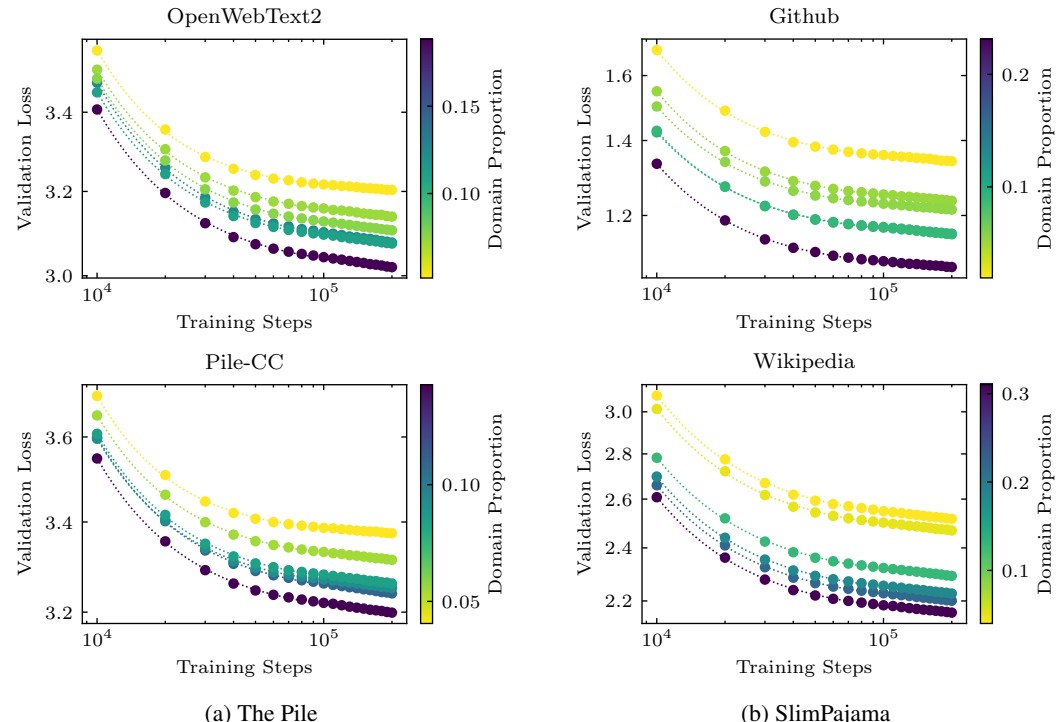

(a) The Pile          (b) SlimPajama

Figure 1: Visualization of the fitting results for Eq. (3) at different domain proportion values, showing the relationship between validation loss and training steps. Each subplot corresponds to a specific domain within different datasets; the points represent the actual observed validation loss, while the dotted lines indicate the fitted results. Both axes are on a logarithmic scale.

curvature. In research related to scaling laws (Kaplan et al., 2020; Hoffmann et al., 2022; OpenAI, 2024), the average validation loss of the model is typically described as following a power law with an irreducible term in relation to the training steps. The decline pattern observed across the various domains aligns with this behavior and the loss for $i$-th domain can be expressed as:

$$L_i(s \mid r_i) = \frac{\tilde{B}_i}{s^{\tilde{\beta}_i}} + \tilde{C}_i. \tag{4}$$

Here, $\tilde{B}_i$ and $\tilde{\beta}_i$ are the scaling factor and exponent of the power-law function, while the additional deviation term $C_i$ is understood as the lower bound for language modeling (Bishop, 2006; Henighan et al., 2020). Further examination of the curves in each subplot reveals that they approximately exhibit a shifting relationship in logarithmic space, described by:

$$\log L_j(s \mid r_j) = \log L_i(s \mid r_i) + \log F_j = \log(L_i(s \mid r_i) \cdot F_j), \tag{5}$$

where $\log F_j$ represents an offset constant. This means that the loss function $L_j$ of $j$-domain can be obtained by multiplying $L_i$ by a scaling factor $F_j$:

$$L_j(s \mid r_j) = L_i(s \mid r_i) \cdot F_j \tag{6}$$

Extending this relationship to all curves within the same domain, the constant $F_j$ can be transformed into a function $f$ related to $r_i$, which is applied to a base scaling function of training steps:

$$L_i(s \mid r_i) = L_{\text{base}}(s \mid r_i) \cdot f(r_i). \tag{7}$$

Combined with Eq. (4), this multiplicative decomposition is clearly associated with Eq. (3), with the following relations: $\tilde{B}_i = B_i \cdot f(r_i)$, $\tilde{C}_i = C_i \cdot f(r_i)$, and $\tilde{\beta}_i = \beta_i$.

**Scaling Domain Proportions Under Fixed Training Steps**      From the other perspective, we analyze how changes in the proportion of a single domain affect its validation loss. The visualization in Fig. 2 follows a similar setting as before. Each line in the figure represents the relationship between

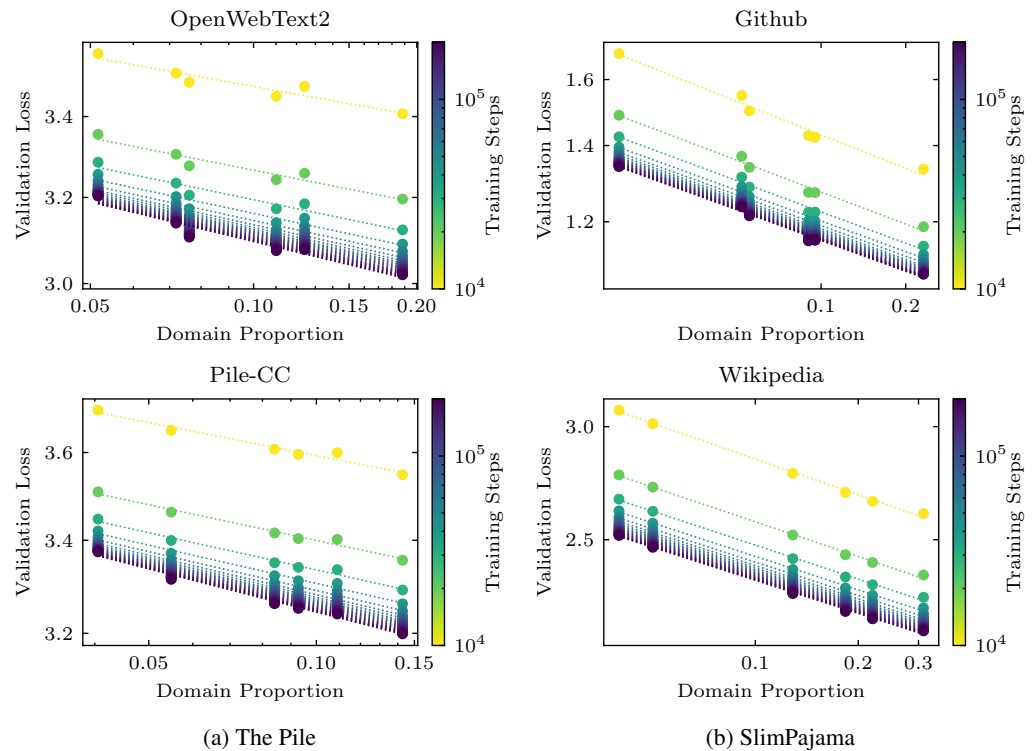

(a) The Pile           (b) SlimPajama

Figure 2: Visualization of the fitting results for Eq. (3) at different numbers of training steps, showing the relationship between validation loss and domain proportion. Each subplot corresponds to a specific domain within different datasets; the points represent the actual observed validation loss, while the dotted lines indicate the fitted results. Both axes are on a logarithmic scale.

validation loss and domain proportion at a specific number of training steps (i.e., training data volume). The points indicate actual observed values, while the dotted line represents the fitted results of Eq. (3). The most notable difference from the previous visualization is the prominent linear relationship observed. This straight line on a logarithmic scale strongly supports the standard power-law function. Thus, the pattern represented by a single straight line in the figure can be expressed as:

$$L_i(r_i \mid s) = \frac{\tilde{A}_i}{r_i^{\tilde{\alpha}_i}}. \tag{8}$$

The collection of straight lines within each subplot can be shifted relative to one another in logarithmic space, leading to the following derivation:

$$\log L_j(r_j \mid s) = \log L_i(r_i \mid s) + \log G_j = \log(L_i(r_i \mid s) \cdot G_j). \tag{9}$$

This implies:

$$L_j(r_j \mid s) = L_i(r_i \mid s) \cdot G_j \tag{10}$$

Considering the unified modeling of these straight lines, the constant $G_j$ can be converted to a function $g$ , yielding the following relationship:

$$L_i(r_i \mid s) = L_{\text{base}}(r_i \mid s) \cdot g(s). \tag{11}$$

Relating this with Eqs. (3) and (9), we obtain the following mappings $\tilde{A}_i = A_i \cdot g(s)$ and $\tilde{\alpha}_i = \alpha_i$.

**Remark** Through the disentanglement of the two input variables, we have identified a separable scaling effect between domain proportions $r_i$ and the number of training steps $s$. Given that dual multiplicative weighting functional forms were derived from both perspectives, we integrated them to construct a mutually modulated bivariate mixing law that encompasses strong interpretability.

## 4 EXPERIMENTAL SETUP

**Datasets**    We employed two well-recognized datasets spanning diverse domains to conduct comprehensive experiments. *The Pile* (Gao et al., 2020) is a diverse language modeling dataset comprising 22 subsets with a total of 825 GiB of textual data. *SlimPajama* (Shen et al., 2023) is a high-quality, seven-domain dataset that has been rigorously deduplicated and refined to 627B tokens from the extensive 1.2T RedPajama dataset (Together Computer, 2023). Following the preprocessing procedures in Xie et al. (2023), we packed all samples within each domain and chunked them into sequences of 1024 tokens for improved training efficiency.

**Model Architecture**    We employed decoder-only transformers based on the DoReMi architecture (Xie et al., 2023). The base model comprises 12 decoder blocks, each with 768-dimensional embeddings, 12 attention heads, and $4\times$ MLP hidden size, matching the DoReMi 280M specifications. For scaled-up experiments on optimized mixtures, we expanded to 16 blocks with 2048-dimensional embeddings and 32 attention heads, aligning with the DoReMi 1B model. All models use the GPT-NeoX tokenizer (Black et al., 2022) with a vocabulary size of 50,277.

**Training Details**    Experiments were conducted under controlled hyperparameters. Each training run consisted of up to 200,000 update steps with a global batch size of 512. The optimizer used was AdamW (Loshchilov & Hutter, 2019) with $\beta_1 = 0.9$, $\beta_2 = 0.99$, $\epsilon = 1 \times 10^{-8}$, and a weight decay of 0.01. The learning rate, initialized at $1 \times 10^{-3}$, decayed exponentially by a factor of $10\times$ over the course of training. We leveraged data parallelism across eight NVIDIA A100 80GB GPUs and bfloat16 mixed precision to improve training throughput. As a reference for the training cost, a single round of DoReMi training took 670 GPU hours on this infrastructure.

**Fitting Details**    As noted previously, we trained models for up to 200,000 update steps on each data mixture, which corresponds to approximately 100 billion tokens. Model evaluation results were collected every 5 billion tokens, allowing for a maximum of 20 assessments, denoted by symbol $n$. During the training of the model on a given data mixture, validation losses for all domains can be obtained simultaneously, with $m$ representing the number of domains. When experiments are conducted on $k$ different data mixtures, we can potentially gather up to $m \times n \times k$ data points in the form of tuples $\langle r_i, s, L_i \rangle$ for fitting **BIMIX**. For the Pile dataset, $m = 22$, while for SlimPajama, $m = 7$. Upon collecting the observational data points, we employ the Trust Region Reflective algorithm (Branch et al., 1999; Virtanen et al., 2020) to fit the coefficients in Eqs. (2) and (3).

**Evaluation Metrics**    The investigated mixing law is fundamentally a specific form of scaling law, widely recognized for its ability to describe the predictability of model loss. Following the prevalent consensus in relevant literature, we primarily report the model's validation loss on each domain, which is sometimes referred to as log-perplexity. To provide a more intuitive understanding of the model's actual performance, we also evaluate model performance on downstream NLP tasks when necessary. The benchmarks from DoReMi are utilized for generative question-answering, specifically WebQuestions (Berant et al., 2013), LAMBADA (Paperno et al., 2016), and TriviaQA (Joshi et al., 2017). We employ the same one-shot prompting and Exact Match metric as used in DoReMi. A response is considered correct if and only if the characters of the model's prediction exactly match those of the True Answer.

**Candidate Mixtures**    Estimating the coefficients in Eqs. (2) and (3) requires a series of observational data points. These are obtained by training on various candidate data mixtures for a limited number of iterations. We consider the following three types of mixtures:

(a) *Baseline*: Represents the original proportions of the datasets, reflecting the intentions of the dataset creators or the inherent distribution of the data collection process.

(b) *DoReMi* (Xie et al., 2023): This approach tunes domain weights by iteratively training reference and proxy models through group distributionally robust optimization. The tuned proportions for the Pile dataset were directly taken from the released results, while for the SlimPajama dataset, we strictly executed the official code to optimize the proportions.

(c) *Entropy*: Measures that serve as efficient proxies for lightweight data mixing, including Shannon entropy (*SE*), conditional entropy (*CE*), joint entropy (*JE*), and von Neumann entropy (*VNE*). Specifically, we calculate the entropy metric for all samples in each domain to represent that domain's data diversity or importance. These entropy values are then normalized across domains to yield mixing proportions that sum to 1. For example, in the case of conditional

entropy, we first tokenize the original dataset, resulting in a token set $\mathcal{D} = (\mathcal{D}_1, \mathcal{D}_2, \ldots, \mathcal{D}_m)$, where each domain $\mathcal{D}_i$ is a set of token sequences $\{(x_1, x_2, \ldots, x_T)\}$ of equal length $T$. The conditional entropy for one domain is computed as follows:

$$H_i(X_i^{(t+1)} \mid X_i^{(t)}) = - \sum_{x \in X_i^{(t)}} \sum_{x' \in X_i^{(t+1)}} P(x, x') \log P(x' \mid x), \tag{12}$$

where $X_i^{(t)}$ and $X_i^{(t+1)}$ are sets of tokens at positions $t$ and $t+1$, respectively. The joint probability $P(x, x')$ and the conditional probability $P(x' \mid x)$ are both statistical estimations on the token set. The mixing proportions $(r_1, r_2, \ldots, r_m)$ are derived by exponentially normalizing the entropy measures:

$$r_i = \frac{e^{H_i}}{\sum_{j=1}^m e^{H_j}}. \tag{13}$$

The resulting proportions place greater emphasis on domains with higher entropy, indicating greater uncertainty, to enhance the learning process. Notably, implementing entropy measurement is highly efficient, as it can be seamlessly integrated into the tokenization process with negligible overhead. Details about the various entropy measures can be found in Appendix A.

## 5 RESULTS AND ANALYSIS

We illustrate the applicability of **BIMIX** from three dimensions: Section 5.1 validates its scalability regarding training data volume; Section 5.2 demonstrates the fitted law's generalization across different mixtures; and Section 5.3 presents a direct method for optimizing domain proportions for application in larger-scale models. Finally, in Section 5.4, we compare data mixtures driven by different entropy measures, offering a streamlined and efficient strategy for data mixing.

### 5.1 EXTRAPOLATING LOSSES ON SCALED-UP DATA

Scaling laws are primarily used to extrapolate model metrics when scaling up the data volume. We held out the validation losses at the final 200,000 steps as the prediction target, and used the remaining observational data points to fit Eqs. (2) and (3). After fitting the coefficients, we used the law to predict the target loss for each domain. Assuming the ground truth loss is $y$ and the **BIMIX**-predicted loss is $y'$, we calculate the relative prediction error as $|y - y'|/y$.

Table 1: Relative prediction error between the real validation loss and the **BIMIX**-predicted validation loss for the final training step. Fitting is performed on all domains for each mixture, reporting the mean, worst, and best errors across domains.

| Dataset | Mixture | Relative Prediction Error ↓ | | |
|---|---|---|---|---|
| | | Mean (%) | Worst (%) | Best (%) |
| The Pile (22 domains) | *Baseline* | 0.16 | 0.43 | 0.03 |
| | *DoReMi* | 0.19 | 0.95 | 0.02 |
| | *CE* | 0.17 | 0.67 | 0.05 |
| SlimPajama (7 domains) | *Baseline* | 0.18 | 0.29 | 0.14 |
| | *DoReMi* | 0.17 | 0.24 | 0.10 |
| | *CE* | 0.18 | 0.31 | 0.12 |

The results are presented in Table 1. It is evident that **BIMIX** extrapolates losses with remarkable accuracy. For both the Pile and SlimPajama datasets, the mean relative error remains below 0.2%. Even for the worst-performing domain, the prediction error is less than 1.0% (DoReMi mixture on the Pile). Moreover, when comparing the error distributions across domains in the two datasets, it is observed that both the worst and best errors on SlimPajama are closer to the mean error, whereas the variance is higher on the Pile. This can be attributed to the greater diversity of the Pile, which comprises up to 22 domains and presents a more significant challenge. In contrast, the meticulous data deduplication applied to SlimPajama helps to reduce cross-domain interference and fitting noise.

## 5.2 ESTIMATING DATA MIXTURES WITHOUT TRAINING

When training language models on multi-source data, determining the appropriate mixing proportions for different domains remains a persistent challenge for practitioners. The naive approach is trial and error, where a few random data mixtures are generated to train models and select the best-performing one. However, as both data and model scales continue to grow, each training session represents a significant expenditure; thus, training large language models has become a careful and strategic process. The proposed **BIMIX** offers a cost-bounded solution based on scaling laws. By training models on a limited number of data mixtures and collecting sufficient observational data points to fit **BIMIX**, we can estimate the effectiveness of any given data mixture without actually conducting training. This approach allows for prospective evaluation of candidate data mixtures before incurring substantial computational costs, helping to eliminate poor options and prioritize effective training configurations.

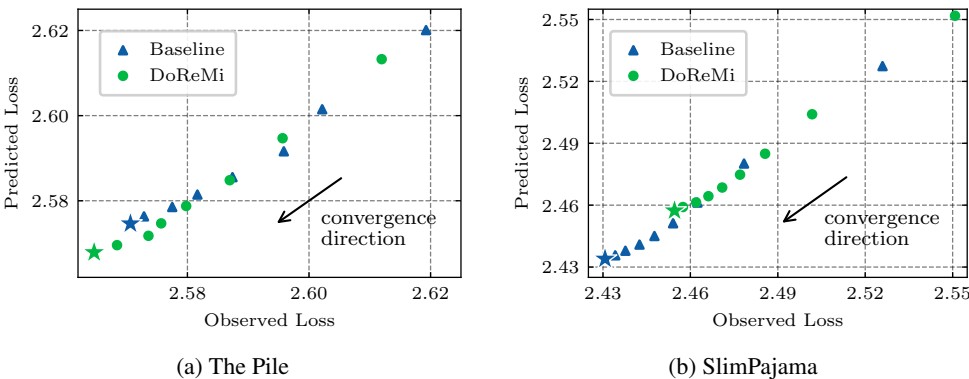

(a) The Pile

(b) SlimPajama

Figure 3: Correlation between the observed validation losses (x-axis) and the **BIMIX**-predicted losses (y-axis) across training iterations with the *Baseline* and *DoReMi* mixtures.

We trained models on the four entropy-driven data mixtures and collected observational data points to fit **BIMIX**. This fitted model was then used to predict the validation losses for each domain at each training step on the *Baseline* and *DoReMi* mixtures. As shown in Fig. 3, the data points from the top right to the bottom left depict the convergence direction as training iterations progress. The x-axis represents the actual loss observed during training, while the y-axis represents the loss predicted by the fitted **BIMIX**. The presence of compact linear trends indicates a strong positive correlation between the two losses. In the lower-left corner of each subplot, the two stars represent the final losses at the end of model training. In Fig. 8a, observing the y-axis reveals that the law predicts the final loss for *DoReMi* to be lower than that for *Baseline*, which is consistent with the actual relative magnitudes displayed on the x-axis; Fig. 8b demonstrates a similar effect.

Table 2: Goodness of fit measured by the coefficient of determination ($R^2$) on validation mixtures. The better the fit, the closer the value is to 1.

| Dataset | Mixture | Goodness of Fit ($R^2$) ↑ | | |
|---|---|---|---|---|
| | | Mean (%) | Worst (%) | Best (%) |
| The Pile | *Baseline* | 0.9748 | 0.7911 | 0.9974 |
| (22 domains) | *DoReMi* | 0.9744 | 0.7864 | 0.9972 |
| SlimPajama | *Baseline* | 0.9940 | 0.9896 | 0.9962 |
| (7 domains) | *DoReMi* | 0.9945 | 0.9904 | 0.9970 |

Table 2 presents a quantitative assessment of the goodness of fit. We employ the commonly used coefficient of determination ($R^2$) (Wright, 1921) metric, which is computed based on the residual sum of squares $SS_{res} = \sum_{j=1}^{n}(y_j - y_j')^2$ and the total sum of squares $SS_{tot} = \sum_{j=1}^{n}(y_j - \bar{y}_j)^2$:

$$R^2 = 1 - \frac{SS_{res}}{SS_{tot}}, \tag{14}$$

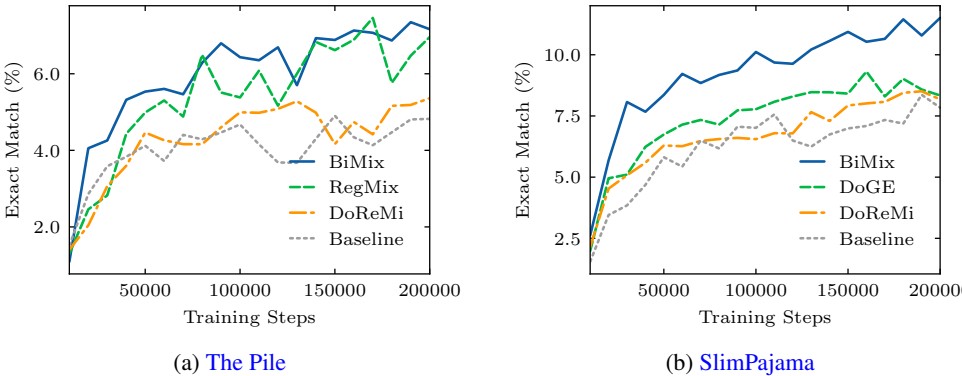

(a) The Pile                    (b) SlimPajama

Figure 4: Comparison of average downstream accuracy of 1B models trained on different data mixtures. Details regarding specific tasks and the Exact Match metric can be found in Section 4.

where $n$ is the number of different training volumes. The $R^2$ value is computed over the sequence of $n$ data points for each domain, on a logarithmic scale to reduce deviation. The better the fit, the closer the value is to 1. Overall, the high mean $R^2$ values suggest that **BIMIX** exhibits good generalization across new data mixtures. The fitted law demonstrates a better fit on the SlimPajama dataset compared to the Pile, as reflected by higher mean and worst $R^2$ values, aligning with the observations made in Section 5.1.

## 5.3 OPTIMIZING DOMAIN PROPORTIONS FOR IMPROVED PERFORMANCE

Recall that **BIMIX** in Eq. (2) describes a system of related equations in which the input variable $\vec{r}$ adheres to a unit-sum constraint. This vectorized formulation facilitates the direct optimization of the proportions across various domains. Consider a common objective of minimizing the model's average validation loss across domains, defined as:

$$\bar{L}(\vec{r}, s) = \sum_{i=1}^{m} L_i(r_i, s),\tag{15}$$

where $L_i$ is the loss function specific to $i$-th domain with proportion $r_i$ and training steps $s$. Solving the following constrained minimization problem yields an optimized domain proportion vector:

$$\vec{r}^* = \operatorname*{arg\,min}_{r_1, r_2, \ldots, r_m} \bar{L}(\vec{r}, s) \quad \text{subject to} \quad \sum_{i=1}^{m} r_i = 1.\tag{16}$$

This presents a classic constrained optimization problem, which can be addressed using Lagrange multipliers and numerical methods (Virtanen et al., 2020). For fitting and optimization, we utilized all observational data points from the four entropy-driven data mixtures, as they serve as effective proxies in practical applications (discussed in Section 5.4).

To validate the efficacy of this approach, we adopt a strategy similar to that in DoReMi (Xie et al., 2023), whereby the optimized mixtures derived from smaller models are utilized to train a larger model with billion-level parameters. Figure 4 illustrates how downstream performance varies as the number of training steps increases. Recent advancements in mixture optimization are also included for comparison, including RegMix (Liu et al., 2024) on the Pile and DoGE (Fan et al., 2023) on SlimPajama. The models trained on the **BIMIX**-optimized mixtures demonstrated performance advantages throughout the training process. While we note that RegMix achieved performance comparable to that of **BIMIX**, it is crucial to highlight that our approach not only optimizes mixtures but also provides a mathematical model for understanding mixing behavior. A detailed comparison of performance across each task is included in Appendix B.

## 5.4 ENTROPY MEASURES AS EFFICIENT MIXING PROXIES

To collect the observational data points necessary for fitting the coefficients of **BIMIX**, we trained models on a series of entropy-driven data mixtures. Entropy essentially quantifies the uncertainty of

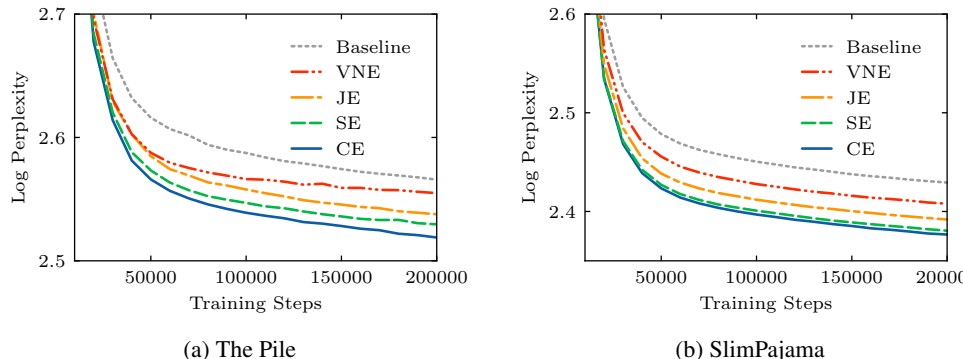

(a) The Pile

(b) SlimPajama

Figure 5: Comparison of log-perplexity evaluations for models trained on different data mixtures.

data distribution, thereby reflecting the learning capacity of domains. It is hypothesized as a valuable proxy. Figure 5 compares the average log-perplexity evaluated on validation sets for models trained on different data mixtures. It is evident that all models trained on these entropy-driven data mixtures exhibit lower log-perplexity compared to the baseline, indicating that the models have learned statistical patterns from the data more effectively. This finding suggests that entropy measures are indeed efficient mixing proxies, facilitating the streamlined initial construction of pretraining datasets. Among these entropy-driven proxies, conditional entropy (CE) consistently demonstrates lower log-perplexity, thereby being regarded as the preferred candidate through experiments.

## 6 DISCUSSION AND FUTURE WORK

Scaling laws model the empirical behavior of model outcomes with respect to certain variables, typically effective within a limited observational range. The applicability of our proposed mixing law under extreme conditions is not guaranteed. In our experiments, domain proportions ranged from 0.0007 to 0.7256, with maximum training data capped at approximately 100B tokens. However, existing research suggests that loss predictability may extend to larger scales (Kaplan et al., 2020; Hoffmann et al., 2022). Our study adheres to settings aligned with relevant work, ensuring consistent domain inclusion during both model training and evaluation (Xie et al., 2023). When evaluating the trained model on new domains, the model's generalization capability and the correlation between new and training domains become primary considerations. This aspect, extending beyond the basic mixing laws studied in this work, warrants dedicated exploration. Our work contributes to the limited research on mixing laws, aiming to establish a foundation for more comprehensive studies.

Reflecting on broader implications, our findings on mixing laws can assist practitioners in optimizing computational resource allocation, promoting advancements in economical and environmentally friendly AI development. This vision is also relevant to the rapidly evolving field of multimodal large models (McKinzie et al., 2024), where processing images, videos, or audio may consume significant computational power. Consequently, exploring the mixing of multimodal training data presents vast opportunities for enhancing model efficiency and performance. Future work could focus on extending our mixing law framework to multimodal contexts, potentially leading to more efficient and effective training paradigms for next-generation AI models.

## 7 CONCLUSION

This paper introduces **BIMIX**, a bivariate data mixing law for language model pretraining. **BIMIX** accurately models the joint scaling behavior of domain proportions and training volume, enabling precise loss extrapolation and generalization to different mixtures. Our experiments demonstrate its effectiveness in optimizing domain proportions, outperforming existing methods. Additionally, we show that entropy-based measures serve as efficient proxies for lightweight data mixing. By offering a nuanced understanding of data mixing dynamics, this research contributes to the development of more efficient large-scale language models and opens avenues for further exploration in data-centric machine learning.

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

# A ENTROPY PROXIES

Given a text dataset such as Pile and SlimPajama, we concatenate all samples within each domain and tokenize them into fixed-length sequences of 1024 tokens. During tokenization, we concurrently record the occurrence frequencies of all unigrams and bigrams, in preparation for computing Shannon entropy, joint entropy, and conditional entropy. The procedure for von Neumann entropy is slightly different: we employ the FastText (Grave et al., 2018) embedding model to map each text sample into a 300-dimensional vector, which will be used to compute their pairwise similarities.

Given a tokenized dataset $\mathcal{D} = (\mathcal{D}_1, \mathcal{D}_2, \ldots, \mathcal{D}_m)$, where each domain $\mathcal{D}_i$ is a set of token sequences $\{(x_1, x_2, \ldots, x_T)\}$ of equal length $T$, the following entropy proxies are computed.

**Shannon Entropy (*SE*)** $H_i(\mathcal{D}_i) = -\sum_{x \in X_i} P(x) \log P(x)$, where $X_i$ is the set of all available tokens in domain $\mathcal{D}_i$ and $P(x)$ denotes the probability of observing token $x$. This proxy quantifies the expected information content associated with token appearances in the dataset, indicative of the corpus diversity.

**Joint Entropy (*JE*)** $H_i(X_i^{(t)}, X_i^{(t+1)}) = -\sum_{x \in X_i^{(t)}} \sum_{x' \in X_i^{(t+1)}} P(x, x') \log P(x, x')$, where $X_i^{(t)}$ and $X_i^{(t+1)}$ represent the sets of tokens at positions $t$ and $t+1$ across all sequences in domain $\mathcal{D}_i$, respectively. The joint probability function $P(x, x')$ was statistically estimated by observing a token $x$ at position $t$, followed by a token $x'$ at position $t+1$. This metric measures the average uncertainty associated with consecutive token pairs and highlights the sequential dependencies in the dataset.

**Conditional Entropy (*CE*)** $H_i(X_i^{(t+1)} \mid X_i^{(t)}) = -\sum_{x \in X_i^{(t)}} \sum_{x' \in X_i^{(t+1)}} P(x, x') \log P(x' \mid x)$, with $X_i^{(t)}$, $X_i^{(t+1)}$, and $P(x, x')$ as previously defined. The term $P(x' \mid x)$ denotes the conditional probability of observing a token $x'$ at position $t+1$ given the presence of token $x$ at position $t$. This measures the anticipated level of surprise when predicting the next token in a sequence, providing a clearer understanding of the text's predictability and its complex linguistic structure.

**Von Neumann Entropy (*VNE*)** In physics, the von Neumann entropy extends the concept of Gibbs entropy from classical statistical mechanics to quantum statistical mechanics. For a quantum-mechanical system described by a density matrix $\rho$, the von Neumann entropy for domain $\mathcal{D}_i$ is defined as

$$H_i(\rho_i) = -\operatorname{Tr}(\rho_i \log \rho_i), \tag{17}$$

where Tr denotes the trace operation and $\log$ the matrix logarithm. Recent research has highlighted its utility in quantifying the diversity of datasets from a system perspective (Friedman & Dieng, 2023). For simplicity, we will drop the subscript $i$ in subsequent discussions, but keep in mind that all discussions pertain to a single domain. In the context of data mixture, we define $\rho$ as $K/N \in \mathbb{R}^{N \times N}$, where $N = |\mathcal{D}_i|$. The matrix $K$ is determined by a positive semi-definite kernel $k : \mathcal{D} \times \mathcal{D} \to \mathbb{R}$, such that $K_{jk} = k(v_j, v_k)$ and $k(v_j, v_k) = 1$ for $1 \leq j, k \leq n$, where $v_j$ and $v_k$ are the embedding vectors processed by FastText. In practice, the von Neumann entropy is computed through the eigenvalues of the density matrix $\rho$:

$$H_i(\rho) = -\sum_{i=1}^{N} \lambda_i \log \lambda_i, \tag{18}$$

where the eigenvalues $\lambda_i$ represent the probability distributions akin to quantum states.

# B    DOWNSTREAM TASK PERFORMANCE

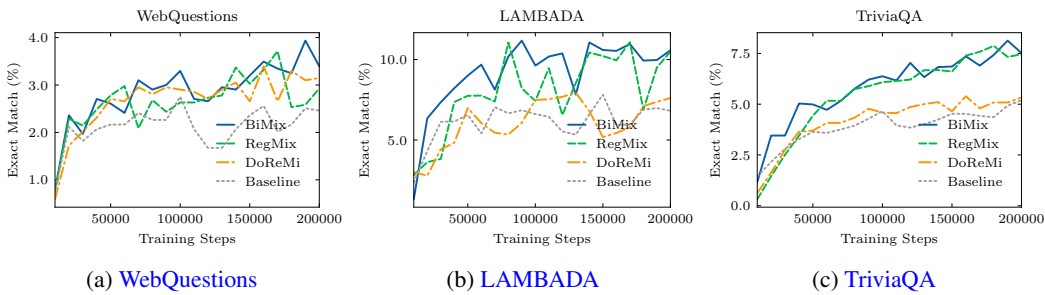

Figure 6: Detailed downstream task performance of the 1B model trained with various data mixtures on the Pile dataset.

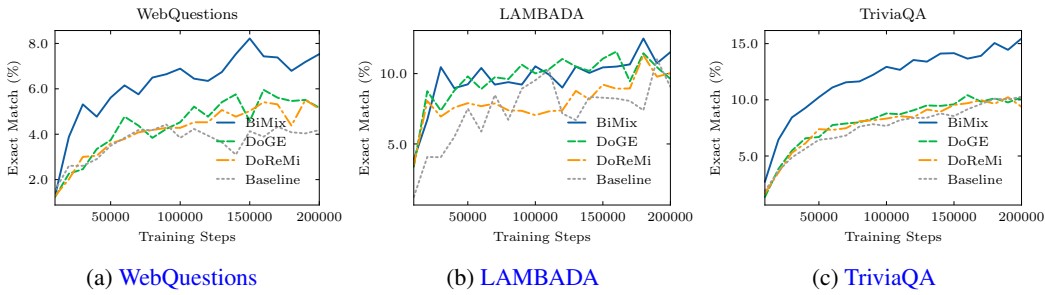

Figure 7: Detailed downstream task performance of the 1B model trained with various data mixtures on the SlimPajama dataset.

Figures 6 and 7 present the downstream performance of the 1B model trained on various data mixtures for the Pile and SlimPajama datasets. Key observations include:

- **BIMIX**-trained models consistently outperform others across most tasks.
- Performance on WebQuestions and LAMBADA shows some variability, but **BIMIX** models maintain overall superiority.
- In TriviaQA, all models demonstrate a stable upward trend with increasing iterations, with **BIMIX** mixtures showing clear advantages.

These results underscore the effectiveness of **BIMIX**-optimized data mixtures in enhancing model performance across diverse downstream tasks.

# C    COMPLEXITY ANALYSIS

Table 3: Complexity comparison between two mixing laws

| Mixing Law | Number of Coefficients | | |
|---|---|---|---|
| | 1 domain 1 target | $m$ domains 1 target | $m$ domains $n$ targets |
| $L_i(r_{1...m}) = c_i + k_i \exp(\sum_{j=1}^{m} t_{ij} r_j)$  (Ye et al., 2024) | $m+2$ | $m^2 + 2m$ | $m^2 n + 2mn$ |
| $L_i(r_i, s) = \frac{A_i}{r_i^{\alpha_i}} \left( \frac{B_i}{s^{\beta_i}} + C_i \right)$  (**BIMIX**) | 2 | $2m$ | $5m$ |

Table 3 compares the fitting complexity of our proposed **BIMIX** with the concurrent composite exponential law by Ye et al. (2024). Both equations model the validation loss for multi-

domain language modeling. The exponential law, $L_i(r_{1...m})$, operates on all domain proportions $(r_i, r_2, \ldots, r_m)$ without considering training steps, while our mixing law incorporates both domain proportion $r$ and training steps $s$. We analyze complexity across three scenarios, progressing from simple to complex.

**Base Case: Fitting an Individual Domain.** The exponential law requires $m + 2$ fitting coefficients, aggregating proportions across $m$ domains with $m$ weighting coefficients $t_{ij}$, plus scaling coefficient $k_i$ and translation coefficient $c_i$. Our bivariate mixing law simplifies to Eq. (8) for fixed training steps, needing only two coefficients: a scaling factor $\tilde{A}_i$ and an exponent $\tilde{\alpha}_i$.

**General Case: Fitting All Domains.** For $m$ domains, both mixing law's coefficient requirement scales by $m$, while our mixing law maintains an order of magnitude fewer coefficients.

**Extensive Case: Fitting Across Multiple Targets.** The exponential law, not accounting for variable $s$, requires $(m^2 + 2m)n$ coefficients for $n$ training step targets. Our bivariate mixing law, incorporating both domain proportions and training steps, shares fitting coefficients across various training steps. Specifically, only five coefficients are required to fit an individual domain across different training steps; this number scales linearly to $5m$ when generalized to all $m$ domains.

Overall, Our bivariate scaling law consistently requires significantly fewer coefficients compared to the composite exponential law in Ye et al. (2024). This reduction translates to fewer required observations for effective fitting, enabling our law to be fit with just a few (potentially as few as two) candidate mixtures, whereas the exponential law needs tens of mixtures. The computational efficiency of our method offers economic and environmental benefits through reduced resource utilization, while simultaneously achieving better data mixtures and enhanced model performance, as demonstrated in Section 5.3.

## D COMPARISION WITH RECENT MIXING LAW

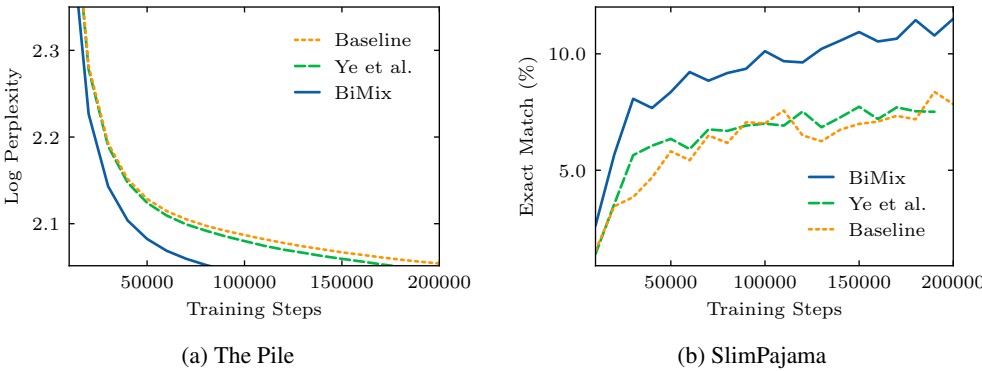

(a) The Pile

(b) SlimPajama

Figure 8: Performance comparison of the 1B model trained on the SlimPajama dataset with recent work by Ye et al. (2024).

We extended the comparison to include the optimal data mixture identified by Ye et al. (2024) for training the 1B model on the SlimPajama dataset. Notably, the exponential law proposed by Ye et al. faces scalability challenges when applied to the more diverse, 22-domain Pile dataset due to the previously discussed quadratic complexity.

Analysis of the results presented in Fig. 8 reveals several key findings:

- Both our **BIMIX**-optimized mixture and Ye et al. (2024)-optimized mixture demonstrated accelerated model convergence compared to the default *Baseline* mixture.

- Our **BIMIX**-optimized data mixture achieved equivalent log-perplexity to the *Baseline* using only 50% of the training steps required by Ye et al. (80,000 vs. 160,000), indicating more effective data utilization.

- While the 1B model trained on the Ye et al.-optimized mixture showed performance comparable to the *Baseline* on downstream tasks, the model trained on our **BIMIX**-optimized mixture exhibited substantial advantages.

These results further underscore the effectiveness of our **BIMIX** approach in optimizing data mixtures for large language model training, offering both improved convergence speed and enhanced downstream task performance.

## E   MIXTURE RECIPES

Tables 4 and 5 provide detailed compositions of the candidate data mixtures employed for the Pile and SlimPajama datasets, respectively. These mixtures form the basis of our experiments and are integral to the evaluation of our **BIMIX** approach.

Table 4: Data mixtures on the Pile dataset

| Domain | *Baseline* | *DoReMi* | *SE* | *CE* | *JE* | *VNE* |
|---|---|---|---|---|---|---|
| ArXiv | 0.0886 | 0.0535 | 0.0336 | 0.0350 | 0.0236 | 0.0258 |
| BookCorpus2 | 0.0044 | 0.0037 | 0.0349 | 0.0461 | 0.0323 | 0.0210 |
| Books3 | 0.0720 | 0.0757 | 0.0544 | 0.0697 | 0.0763 | 0.0283 |
| DM Mathematics | 0.0204 | 0.0019 | 0.0047 | 0.0078 | 0.0007 | 0.0434 |
| Enron Emails | 0.0029 | 0.0040 | 0.0416 | 0.0272 | 0.0228 | 0.0717 |
| EuroParl | 0.0075 | 0.0120 | 0.0681 | 0.0314 | 0.0431 | 0.0709 |
| FreeLaw | 0.0403 | 0.0380 | 0.0375 | 0.0394 | 0.0297 | 0.0334 |
| GitHub | 0.0554 | 0.0325 | 0.0382 | 0.0451 | 0.0346 | 0.1021 |
| Gutenberg (PG-19) | 0.0218 | 0.0292 | 0.0348 | 0.0535 | 0.0375 | 0.0211 |
| HackerNews | 0.0079 | 0.0084 | 0.0343 | 0.0474 | 0.0328 | 0.0404 |
| NIH ExPorter | 0.0047 | 0.0084 | 0.0437 | 0.0406 | 0.0357 | 0.0321 |
| OpenSubtitles | 0.0110 | 0.0032 | 0.0161 | 0.0202 | 0.0065 | 0.0202 |
| OpenWebText2 | 0.1242 | 0.1905 | 0.0719 | 0.0761 | 0.1101 | 0.0517 |
| PhilPapers | 0.0032 | 0.0093 | 0.0772 | 0.0603 | 0.0936 | 0.0390 |
| Pile-CC | 0.1090 | 0.1379 | 0.0549 | 0.0841 | 0.0928 | 0.0405 |
| PubMed Abstracts | 0.0756 | 0.0970 | 0.0541 | 0.0497 | 0.0541 | 0.0423 |
| PubMed Central | 0.1139 | 0.0608 | 0.0536 | 0.0462 | 0.0498 | 0.0383 |
| StackExchange | 0.0907 | 0.0746 | 0.0431 | 0.0506 | 0.0439 | 0.0428 |
| USPTO Backgrounds | 0.0401 | 0.0327 | 0.0362 | 0.0423 | 0.0308 | 0.0321 |
| Ubuntu IRC | 0.0098 | 0.0083 | 0.0374 | 0.0281 | 0.0212 | 0.0529 |
| Wikipedia (en) | 0.0894 | 0.1068 | 0.0626 | 0.0639 | 0.0804 | 0.0784 |
| YoutubeSubtitles | 0.0072 | 0.0117 | 0.0670 | 0.0353 | 0.0476 | 0.0716 |

Table 5: Data mixtures on the SlimPajama dataset

| Domain | *Baseline* | *DoReMi* | *SE* | CE | *JE* | *VNE* |
|---|---|---|---|---|---|---|
| ArXiv | 0.0458 | 0.0213 | 0.0738 | 0.0710 | 0.0339 | 0.0598 |
| Books | 0.0420 | 0.0420 | 0.1474 | 0.1669 | 0.1592 | 0.0700 |
| C4 | 0.2660 | 0.2864 | 0.1619 | 0.2079 | 0.2179 | 0.1265 |
| CommonCrawl | 0.5203 | 0.5518 | 0.1863 | 0.2175 | 0.2623 | 0.0933 |
| GitHub | 0.0522 | 0.0191 | 0.0952 | 0.0902 | 0.0556 | 0.2314 |
| StackExchange | 0.0337 | 0.0292 | 0.1152 | 0.1177 | 0.0878 | 0.1088 |
| Wikpedia | 0.0399 | 0.0501 | 0.2202 | 0.1287 | 0.1834 | 0.3103 |

