# OpenReview forum: "BiMix: Bivariate Data Mixing Law for Language Model Pretraining"
_ICLR.cc/2025/Conference — Submitted to ICLR 2025_

### Official Review · Reviewer_EKGu · 2024-10-30

**Soundness:** 2
**Presentation:** 2
**Contribution:** 2
**Rating:** 3
**Confidence:** 4

**Summary:**

The paper introduces BiMix, a bivariate data mixing framework designed to optimize and understand the impact of multi-domain data in LLM pre-training.
BiMix models the joint scaling behavior of domain proportions and data volume to improve data mixing in LLM pre-training.
Key contributions include:
1. A mathematical framework that models domain proportions and data volume scaling to enhance interpretability in data mixing.
2. Experimental validation demonstrating BiMix's ability to predict and optimize model performance across diverse data mixtures, outperforming traditional approaches.
3. Introduction of entropy-based measures as efficient, computationally lightweight proxies for data mixing, facilitating faster and more effective pretraining.

**Strengths:**

1. The research topic on pre-training data mixture is timely and important for the LLM community.

2. The evaluation is relatively comprehensive. Experiments on two large-scale datasets (The Pile and SlimPajama) demonstrated BiMix's high accuracy in loss extrapolation (mean relative error <0.2%) and its generalization to unseen mixtures (R2 >0.97). Optimizing data mixtures based on the proposed method yields superior model performance compared to existing methods, in terms of both convergence speed and downstream task performance.

3. The proposed entropy-based measures help to reduce the computational overhead of searching data mixtures.

**Weaknesses:**

1. The validness of the underlying domain-independent assumption. Similar to other recent papers that model the data mixture problem using scaling law like [1], there is also an implicit assumption of the proposed method: different domains are independent. In Eq. (3), no interaction term models the interactions between different domains. We would like to understand how each domain affects every other domain, this interaction might be highly complex and not local.

2. Missing baselines. There does not seem to be a sufficient comparison between BiMix and other data selection methods, e.g. comparing with other token-level and sample-level automatic data selection methods [2, 3].

3. The claim of *theoretical insights into data mixing dynamics* seems to be an overclaim because there is no theoretical justification.

[1] Data Mixing Laws: Optimizing Data Mixtures by Predicting Language Modeling Performance

[2] Perplexed by Perplexity: Perplexity-Based Data Pruning With Small Reference Models

[3] DoGE: Domain Reweighting with Generalization Estimation

**Questions:**

1. The design choice of different entropy-based measures and how to derive the mixing proportions from them are hard to follow.
Could the authors please elaborate more on this?

2. As shown in Figure 6, the downstream task performances of the model trained by different mixtures are fairly low, e.g. mostly under 10%. Are the models significantly better than random guesses and how is the variance of the downstream task performances?

**Details Of Ethics Concerns:**

The Pile dataset used in this paper is the 22-domain version, which has copyright issues as discussed in https://huggingface.co/datasets/monology/pile-uncopyrighted.

---

> ### Author Response · Authors · 2024-11-28
> **Response to Reviewer EKGu (part 1/2)**
>
> We would like to express our sincere gratitude for your comments and queries. Below, we address your feedback regarding Weaknesses (W) and Questions (Q) point by point, and we have also included additional experiments in response to your suggestions.
>
> ---
>
> **(W1) The validness of the underlying domain-independent assumption.**
>
> We would like to clarify that the domain-independent assumption is not arbitrary; it is grounded in observations of real-world scaling behaviors, which we elaborate upon in Section 3.2. In this subsection, we fixed one of the independent variables (either training steps or domain proportions) and examined the scaling relationship between validation loss and the other variable. We find that the resulting scaling patterns could be effectively modeled using a domain-independent approach. This finding is supported by the visualizations in Figures 1 and 2, as well as experimental results in Sections 5.1 and 5.2.
>
> Methods such as those proposed by Ye et al. [1] employ domain-dependent modeling approaches, which we have shown to involve redundant parameters that adversely affect their practicality. Through theoretical analysis and experimental validation, we confirm the advantages of our modeling in terms of both efficiency and performance:
>
> - **Computational Complexity**: In Appendix C, we analyzed the relationship between the number of parameters required for fitting the two mixing laws and the number of domains. Our method clearly exhibits a complexity that is `an order of magnitude lower` than that of Ye et al. This significantly affects the scalability of Ye et al.'s approach when dealing with multiple domains. For example, if we consider mixing 10 data domains ($m=10$), Ye et al.'s mixing law requires fitting 120 parameters, while our BiMix only requires 20 parameters. Consequently, Ye et al.'s method would necessitate training hundreds of models to gather sufficient observation data points for fitting, which is often impractical.
>
> - **Mixture Optimization**: In Appendix D, we compared the performance of models trained on the mixtures optimized by the two mixing laws. Our BiMix-driven models not only achieve faster convergence (requiring only 50\% of the iterations to reach comparable levels) but also demonstrate higher accuracy in downstream task evaluations.
>
> Our findings reveal an intriguing phenomenon: the interference between domains may not be as critical as previously assumed, provided that strict deduplication among domains is maintained. The output of SlimPajama involved rigorous deduplication across the RedPajama domains, thus aligning with our expectations. Additionally, as noted in Table 2, the prediction error of the law on the Pile dataset is slightly larger compared to SlimPajama, attributed to less effective deduplication among the Pile domains.
>
>
> **(W2) Missing baselines.**
>
> We have carefully examined the methodologies in the two papers you mentioned. However, the code related to "Perplexed by Perplexity" [2] is not publicly available, so we are only able to include experiments based on the DoGE [3] method. We have updated the comparison results in Figures 4(b) and 7, which continue to demonstrate the advantages of our approach.
>
> **(W3) The claim of theoretical insights into data mixing dynamics seems to be an overclaim because there is no theoretical justification.**
>
> We apologize for any misunderstanding caused by the language used. Our intent was to convey that the proposed BiMix equation mathematically describes the relationship between validation loss, training steps, and domain proportions. To avoid any potential confusion, we have removed the term "theoretical" to provide a more precise expression.

---

> ### Author Response · Authors · 2024-11-28
> **Response to Reviewer EKGu (part 2/2)**
>
> **(Q1) The design choice of different entropy-based measures and how to derive the mixing proportions from them are hard to follow. Could the authors please elaborate more on this?**
>
> In Appendix A, we have provided the specific formulas for the four entropy-based mixtures. Here is a more detailed explanation of the entire process:
>
> 1. During model training, we need to sample a batch that includes data from different domains, which requires determining the proportion of each domain.
> 2. Entropy is commonly understood as a measure of the inherent information contained within data. In probability theory, higher information content indicates greater uncertainty. A domain with higher information content/uncertainty is more challenging to learn and thus requires a larger proportion in the mixture.
> 3. For each entropy definition employed, we count the frequency of tokens for each domain and compute the overall information represented by that domain using mathematical formulas.
> 4. Finally, we normalize the entropy values of each domain using the softmax function (as shown in Equation 13) to obtain proportions that sum to one, thereby generating the corresponding mixture.
>
>
> **(Q2) As shown in Figure 6, the downstream task performances of the model trained by different mixtures are fairly low，e.g. mostly under 10\%.**
>
> This result is expected. We utilized the evaluation code provided by DoReMi and achieved results comparable to those reported in their original paper. For instance, in Table 5 of DoReMi Appendix B, even an 8B model achieves an accuracy of only 4.35\% on NaturalQuestions and 6.74\% on WebQuestions. It is important to note that these results were obtained using the Exact Match evaluation metric and a one-shot prompt setting. You may have observed higher performance in other papers, but discrepancies in metrics and few-shot settings across different studies make such comparisons inadvisable.
>
> ---
>
> [1] Data Mixing Laws: Optimizing Data Mixtures by Predicting Language Modeling Performance
>
> [2] Perplexed by Perplexity: Perplexity-Based Data Pruning With Small Reference Models
>
> [3] DoGE: Domain Reweighting with Generalization Estimation
>
> ---
>
> We hope that our responses have effectively addressed your concerns and you will consider re-evaluating our paper favorably. If you have any further suggestions or questions, please feel free to contact us. Thank you once again for your time and effort.

---

> ### Author Response · Authors · 2024-12-04
>
> Dear Reviewer EKGu,
>
> We hope this message finds you well. We have thoroughly addressed each of the weaknesses and questions you raised, and as per your suggestions, we have included additional experiments with more baselines. As the deadline approaches, we wanted to check in to ensure that our rebuttal thoroughly addresses your concerns.
>
> If you find our responses helpful, we sincerely request that you consider increasing your score accordingly and take our efforts and improvements into account in your final decision.
>
> Thank you once again for your invaluable time and effort. Your positive evaluation and constructive feedback are immensely important to us.
>
> Best regards,
> Authors

---

### Official Review · Reviewer_VKW2 · 2024-11-02

**Soundness:** 3
**Presentation:** 3
**Contribution:** 3
**Rating:** 5
**Confidence:** 3

**Summary:**

This paper introduces BIMIX, a data mixing law for better understanding and optimizing data mixtures across different domains.
The key contributions of the paper are:
1. They provides a scalable framework to model the relationship between domain proportions and training data quantity.
2. Through experiments on two large-scale datasets, Pile and SlimPajama, BIMIX predicts model loss with a mean relative error below 0.2%.
3. The authors introduce entropy-based measures as efficient and computationally lightweight proxies for data mixing, offering a practical alternative to resource-intensive optimization methods.

**Strengths:**

1. BIMIX is a novel methods, proposing a bivariate data mixing law with models domain proportions and training data quantity together. They also incorporates entropy-based measures as efficient proxies to fastly optimizate the training.
2. By validating BIMIX with validation error, the paper demonstrates robustness, presenting a comprehensive evaluation for practical effectiveness and applicability.
3. The paper is well-organized, effectively explaining complex ideas with clear definitions.

**Weaknesses:**

1. The downstream evaluation includes only PPL. This limits the understanding of how BIMIX-optimized mixtures perform on other important tasks, such as ARC, hellaswag, pica, openbookqa, boolq and so on.
2. While the authors aim to reduce computational costs, the paper lacks a detailed comparison of its efficiency. For example, how much GPU hours do we need in order to get the good mixture.
3.  The paper primarily contrasts BIMIX with DoReMi and baseline mixtures, without comparisons to other recent advancements, such as REGMIX[1] or Online Data Mixing[2].

1. RegMix: Data Mixture as Regression for Language Model Pre-training
2. Efficient Online Data Mixing For Language Model Pre-Training

**Questions:**

I have a little bit confused about the part 5.4 **ENTROPY MEASURES AS EFFICIENT MIXING PROXIES**, it seems that the autor trained
models on a series of entropy-driven data mixtures in order to collect the observational data points necessary for fitting the coefficients of BIMIX. After fitting the BIMIX, we can final get the final mixture for better pretraining. However, at the end of part 5.4, the author seems to show that entropy measures mixing proxy is already enough and do not talk much about the BIMIX.

---

> ### Author Response · Authors · 2024-11-28
>
> We sincerely appreciate your acknowledgment of our work and the valuable feedback you provided. Below, we address the identified Weaknesses (W) and Questions (Q) point by point and includes additional experiments in accordance with your suggestions.
>
> ---
>
> **(W1) The downstream evaluation includes only PPL.**
>
> In addition to PPL, we have incorporated evaluations on downstream tasks such as WebQuestions, LAMBADA, and TriviaQA, maintaining consistency with the DoReMi paper. These results are presented in Figures 4, 6, 7, and 8(b).
>
>
> **(W2) While the authors aim to reduce computational costs, the paper lacks a detailed comparison of its efficiency. For example, how much GPU hours do we need in order to get the good mixture.**
>
> In Appendix C, we have provided a comprehensive comparison of the fitting complexity between our method and that of Ye et al. [1] The analyses indicate that our approach is an order of magnitude more efficient. It is important to clarify that once the mixing law is fitted, it no longer relies on GPU resources; instead, mixture optimization can be achieved with just a few seconds of CPU computation.
>
> **(W3) The paper primarily contrasts BIMIX with DoReMi and baseline mixtures, without comparisons to other recent advancements, such as REGMIX [2] or Online Data Mixing [3].**
>
> We have carefully examined the methodologies in these two papers and have included a comparison with RegMix in Figures 4(a) and 6. While we observe that it demonstrates comparable performance, it is crucial to note the differences and emphasize that the focus of our study is on the mathematical explainability of mixing behavior. We have proposed a bivariate mixing law and illustrated its broader applicability, in contrast to RegMix, which primarily concentrates on optimizing mixtures. Additionally, the Online Data Mixing approach employs a fundamentally different online learning framework, which we were unable to adapt to our framework within the limited time available for this rebuttal.
>
>
> **Questions about section 5.4**
>
> The primary objective of the entropy-based mixtures is to gather the necessary data points for fitting our law. However, given the significant computational resources required for training language models and studying scaling laws, we aim to maximize resource utilization to extract valuable insights during this process. In Section 5.4, we contribute our findings to the community to assist practitioners by demonstrating that entropy-driven data mixtures can serve as an efficient alternative. While they may be slightly inferior to the fitted mixing law, they offer a solid starting point at a significantly lower computational cost. We believe this has positive implications for the "data mixing" community, particularly in resource-constrained scenarios.
>
> ---
>
> [1] Data Mixing Laws: Optimizing Data Mixtures by Predicting Language Modeling Performance
>
> [2] RegMix: Data Mixture as Regression for Language Model Pre-training
>
> [3] Efficient Online Data Mixing For Language Model Pre-Training
>
> ---
>
> We sincerely hope that our responses adequately address your comments and encourage you to consider a favorable acceptance of our paper. Please do not hesitate to contact us if you have any further questions. Thank you once again for your helpful assessment!

---

> ### Author Response · Authors · 2024-12-04
>
> Dear Reviewer VKW2,
>
> We greatly appreciate your time in reviewing our work and truly value your thoughtful consideration. We have thoroughly addressed the misunderstandings you initially identified. Specifically, we clarified that our evaluation is not limited to PPL but also includes downstream tasks (**W1**), and we have detailed in the appendix how both theoretical and experimental evidence support the efficiency of our method (**W2**). In response to your suggestions, we have also conducted more experimental comparisons (**W3**). Furthermore, we have provided detailed explanations to resolve your concerns regarding Section 5.4.
>
> We have made every effort to address each of your comments and sincerely hope that you will consider these improvements when making your final decision. If you have any further questions or concerns, please do not hesitate to let us know, and we will do our utmost to respond promptly.
>
> Best regards,
> Authors

---

### Official Review · Reviewer_oXCG · 2024-11-03

**Soundness:** 2
**Presentation:** 2
**Contribution:** 2
**Rating:** 5
**Confidence:** 5

**Summary:**

This paper introduces a bivariate data mixing law for LLM pretraining, which disentangles models domain proportions and training steps and establishes a connection between domain validation loss and these variables. Empirical experiments validate this proposed scaling law.

For the experiments, first, this paper assesses the extrapolated loss prediction over full training steps, showing minimal prediction error. Second, four entropy-based mixture candidates are trained to fit the mixing law, and these results are used to predict the loss for two other data mixtures (baseline and doremi) without actual training, demonstrating a strong fit. Third, once the scaling law is established, this paper performs constrained optimization to determine optimal domain proportions, outperforming baseline methods in three generative question-answering tasks. Finally, the entropy-based mixtures are shown to consistently perform better than the baseline.

**Strengths:**

- The disentanglement of domain proportions and training steps is novel and interesting.
- The experimental results in Sections 3.2, 5.1, and 5.2 are promising, indicating that the proposed formulation is both effective and sound.
- This paper is generally clear and easy-to-follow, addressing the significant problem of data mixing in LLM pretraining.

**Weaknesses:**

The primary concern with this paper lies in the empirical examination and application of the proposed data-mixing law. Section 5.2 suggests that the data-mixing law can estimate the impact (loss) of certain data mixture proportions without actual training. However, only two data mixture proportions (baseline and doremi) are tested, which may not provide sufficient evidence of general applicability.  Furthermore, these two proportions are highly similar (see Appendix D, where the correlation is approximately 0.9), which further limits the effectiveness of the verification. While I acknowledge the significant costs associated with additional testing, a more comprehensive validation of the proposed law would strengthen its credibility.

Section 5.3 examines the optimized proportion based on the scaling law but is limited to only three generative question-answering tasks, which is insufficient to conclusively demonstrate the optimality of this proportion. This aspect is crucial, as it represents a key use case for the proposed law.

Additionally, the presentation could be improved. Section 3 would benefit from reorganization to enhance clarity. In Sections 5.1 to 5.3, it would also be helpful to specify how many and which data points (i.e., at what training steps and with what proportion ratios) were used to fit the scaling law and predict specific points (i.e., at what training steps and with what proportion ratios). This information is crucial for assessing the practical utility of the scaling law.

**Questions:**

1. Could you clarify the logical flow between Sections 3.1 and 3.2? My understanding is that Equation (3) is initially proposed, followed by two empirical experiments in Section 3.2 aimed at verifying the proposed law. If this is correct, I suggest avoiding the term "defined" in Line 135, as it may cause confusion.
2. In Sections 5.1 to 5.3, could you specify **how many** and **which** (i.e., at what training steps and with what proportion ratios) data points were collected to *fit* the scaling law and *predict* **which** target data point (i.e., at what training steps and with what proportion ratios)? Presenting this information clearly would be essential to assess the scaling law's utility.
3. Based on my understanding, the four entropy-based data mixtures are used to fit the data mixing laws.
- To fit a reliable function, it might be preferable to use more scattered-out data points rather than closely related ones.  Why not employ more diverse proportions rather than relying solely on entropy-based methods, which could be quite similar (as indicated by the correlations)?
- Could you clarify the purpose of Section 5.4? If the entropy-based methods are only used to gather observational data points, what is the significance of demonstrating their superiority over the baseline? Should this be considered an incidental result, or does it have a more central role in validating the proposed method?
4. The introduction (Line 59) claims improved convergence speed. Could you provide concrete evidence to support this?
5. What is the log perplexity of doremi? This result seems to be missing from both Section 5.3 and Figure 8(a) in Appendix D.
6. Beyond optimizing data proportions, what other practical use cases could the proposed law have? If this optimization is the primary use case, I would recommend placing greater emphasis on Section 5.3.

Minor suggestions (not affecting scores):
1. $s$ in Equation (2) can be explicitly defined.
2. It may be beneficial to compare recent related literature:

[1] Kang, Feiyang, Yifan Sun, Bingbing Wen, Si Chen, Dawn Song, Rafid Mahmood, and Ruoxi Jia. "Autoscale: Automatic prediction of compute-optimal data composition for training llms." arXiv preprint arXiv:2407.20177 (2024).

[2] Liu, Qian, Xiaosen Zheng, Niklas Muennighoff, Guangtao Zeng, Longxu Dou, Tianyu Pang, Jing Jiang, and Min Lin. "Regmix: Data mixture as regression for language model pre-training." arXiv preprint arXiv:2407.01492 (2024).

I am willing to raise the score if the authors can sufficiently address the above questions.

---

> ### Author Response · Authors · 2024-11-28
> **Response to Reviewer oXCG (part 1/2)**
>
> We sincerely appreciate the valuable time you have dedicated to reviewing our work and thank you for your insightful feedback. We address your questions one by one and hope that you will consider re-evaluating our work.
>
> ---
>
> **(Q1) The logical flow between Sections 3.1 and 3.2.**
>
> In Section 3.1, we prominently present our proposed mixing law to provide clarity and prevent readers from getting lost in complex derivations. Section 3.2, on the other hand, offers a detailed discussion of the interpretability of this modeling rather than empirical experiments. This subsection systematically unfolds our observations, thoughts, and the inductive process leading to BiMix, effectively demonstrating that our proposed equation is a credible description of the scaling behavior rather than being an arbitrary or baseless claim.
>
> **(Q2) In Sections 5.1 to 5.3, how many and which data points were collected to fit the scaling law and predict which target data point?**
>
> As you correctly summarized in Q3, we collect data points from four entropy-based data mixtures to fit the mixing law, as indicated in Lines 407 and 471 of the manuscript. The specific proportion ratios of these mixtures are presented in Tables 4 and 5 of Appendix E.
>
> The calculation of the number of data points is detailed in the “Fitting Details” paragraph (Lines 292 to 299), where the formula is provided as $m \times n \times k$. Here, $m$ represents the number of domains (22 for Pile and 7 for SlimPajama), $n$ is the number of evaluations (20), and $k$ is the number of mixtures. Due to different experimental setups across the three sections, the specific data points used vary slightly:
>
> - In Section 5.1, the focus is on the law's extrapolation capabilities concerning validation loss. For each mixture in Table 1, we hold out the loss at the maximum training step of 200,000 as the validation set, while prior data points are used to fit the law. Consequently, the number of data points used to fit the law is $22 \times (20 - 1)$ for Pile and $7 \times (20 - 1)$ for SlimPajama; the remaining 22 for Pile and 7 for SlimPajama serve as the target predictions.
>
> - In Section 5.2, we assess the law’s generalization to new mixtures. We use all data points from the four entropy-based data mixtures to fit the law, totaling $22 \times 20 \times 4$ for Pile and $7 \times 20 \times 4$ for SlimPajama. The target predictions come from the data points collected during training on the other two mixtures (Baseline and DoReMi), specifically $22 \times 20 \times 2$ for Pile and $7 \times 20 \times 2$ for SlimPajama. For visual clarity, we downsampled the target data points in Figure 3 during presentation.
>
> - In Section 5.3, we also used all data points from the four entropy-based data mixtures to fit the law (there was no need to set aside target data points). The fitted law was subsequently utilized to derive optimized domain proportions.
>
>
> **(Q3) Clarification on entropy-based data mixtures:**
>
> **(Q3.1) Why not employ a more diverse range of proportions rather than relying solely on entropy-based methods?**
>
> We computed the Pearson correlation coefficients among the four entropy-based mixtures. Notably, aside from the correlation between CE and JE, which is only slightly above 0.9, the correlations among SE, CE, JE, and VNE are relatively dispersed. In contrast, DoReMi and Baseline exhibit a correlation of 0.8882, while the correlations among our four entropy-based mixtures are almost all below this value.
>
> |         | SE | CE | JE | VNE |
> |:---------|---:|---:|---:|----:|
> | SE       | 1.0000 | 0.6332 | 0.8357 | 0.3382 |
> | CE       |  | 1.0000 | 0.9011 | -0.0380 |
> | JE       |  |  | 1.0000 | 0.1270 |
> | VNE      |  |  |  | 1.0000 |
>
>
> **(Q3.2) The purpose of Section 5.4.**
>
> The use of entropy-based mixtures serves two key purposes. First, they are employed to gather the necessary data points for fitting our law. Second, we aim to extract valuable insights during this process. Given the significant computational resources required for training language models and studying scaling laws, we preferred not to waste resources on random mixtures. In Section 5.4, we contribute our findings to the community: entropy-driven data mixtures can serve as an efficient alternative that, while potentially slightly inferior to the fitted mixing law, offers a solid starting point at a significantly lower computational cost, especially in resource-constrained scenarios.

---

> ### Author Response · Authors · 2024-11-28
> **Response to Reviewer oXCG (part 2/2)**
>
> **(Q4) The introduction (Line 59) claims improved convergence speed. Could you provide concrete evidence to support this?**
>
> Our conclusion regarding faster convergence is based on the analysis presented in Figures 5 and 8(a), where models trained on CE or BiMix mixtures exhibited a more rapid decrease in validation loss compared to those trained using other mixtures. For example, as shown in Figure 8(a), our BiMix-optimized data mixture achieved the same log-perplexity as the Baseline using only 50\% of the training steps (80,000 vs. 160,000) required by Ye et al. [1]
>
> **(Q5) What is the log perplexity of DoReMi?**
>
> The log perplexity values are equivalent to the validation loss; the former is simply a different nomenclature for the evaluation perspective.We chose to present the results this way to maintain consistency with the metrics used in the DoReMi paper and to avoid any potential misunderstandings  related to the evaluation metrics. We will clarify this point in the manuscript.
>
> **(Q6) Beyond optimizing data proportions, what other practical use cases could the proposed law have?**
>
> We have illustrated three practical applications of the proposed law in Sections 5.1 to 5.3. Section 5.1 showcases a fundamental application of scaling laws by allowing us to make reasonable predictions about the model's future performance based on current observations before committing additional computational resources. Sections 5.2 and 5.3 explore further applications directly related to "mixing"—one focuses on predicting the model's performance under novel data mixtures, while the other aims at optimizing the mixture proportions directly.
>
>
> **Minor Suggestions:**
>
> 1. We explicitly defined $s$ in Equation (2) within the manuscript.
> 2. We have reviewed the literature you recommended and added comparisons to RegMix [2] and DoGE [3] in Figure 4 and Section 5.3.
>
> ---
>
> [1] Data Mixing Laws: Optimizing Data Mixtures by Predicting Language Modeling Performance
>
> [2] Regmix: Data mixture as regression for language model pre-training
>
> [3] DoGE: Domain Reweighting with Generalization Estimation
>
> ---
>
> We hope that our responses have adequately addressed your concerns and encourage you to consider raising the score. If you have any additional questions or need further clarification, please do not hesitate to reach out. We greatly value your time and effort and are grateful for your support.

---

> > ### Comment · Reviewer_oXCG · 2024-11-29
> >
> > Thank you for your detailed responses. I have a few follow-up suggestions and observations:
> >
> > 1. Please ensure that Q1, Q2, Q3.2, and Q5 are clearly clarified in the manuscript (or appendix) for better readability and transparency.
> > 2. For Q3.1, based on the correlation analysis you showed, SE, CE, and JE already show high correlations, which does not fully address the question about exploring diverse alternatives to entropy-based mixtures.
> > 3. Unfortunately, I did not see a direct response to the weaknesses I highlighted in my initial review. Without a proper response of these concerns, they remain significant and could undermine the overall contribution and impact of the work.

---

> > > ### Author Response · Authors · 2024-11-30
> > > **Thanks for your prompt reply!**
> > >
> > > Thank you for your continued feedback! We have carefully considered your comments and provide responses below to directly address all the raised weaknesses:
> > >
> > > ---
> > >
> > > ## Weaknesses Paragraph 1 (W1)
> > >
> > > 1. Regarding your comments in **W1**, as well as the follow-up to **Q3.1**, we have identified two concerns related to the experiments in Section 5.2:
> > >
> > >    - Testing on only two mixtures may not provide sufficient evidence for general applicability.
> > >    - Fitting with four entropy-based mixtures may lack diversity, primarily due to high correlations among them.
> > >
> > >    As per your suggestion, "*a more comprehensive validation of the proposed law would strengthen its credibility.*" To address these concerns, we applied the well-established approach `Leave-One-Out Cross-Validation` (also equivalent to `6-fold cross-validation` in this context) to offer a reliable and unbiased estimate of model generalization. In each iteration, one mixture is held out for prediction, while the remaining mixtures are used for fitting. This process was repeated six times, providing `comprehensive validation` and accounting for `various correlation combinations`. The tables below present the improved Goodness-of-Fit results and the mean cross-correlation between the held-out and fitted mixtures.
> > >
> > >    Table 1: Leave-One-Out Cross-Validation on SlimPajama
> > >
> > >    | Hold-out Mixture | Goodness of Fit ($R^2$) | Cross Correlation |
> > >    |------------------|------------------------|------------------|
> > >    | Baseline         | 0.9918                 | 0.5417           |
> > >    | DoReMi           | 0.9982                 | 0.5715           |
> > >    | SE               | 0.9957                 | 0.5732           |
> > >    | CE               | 0.9971                 | 0.6007           |
> > >    | JE               | 0.9881                 | 0.6805           |
> > >    | VNE              | 0.9528                 | -0.0489          |
> > >    | Stats (mean$\pm$std) | 0.9873$\pm$0.0158  |                  |
> > >
> > >    Table 2: Leave-One-Out Cross-Validation on the Pile
> > >
> > >    | Hold-out Mixture | Goodness of Fit ($R^2$) | Cross Correlation |
> > >    |------------------|-------------------------|------------------|
> > >    | Baseline         | 0.9778                  | 0.4713           |
> > >    | DoReMi           | 0.9793                  | 0.5840           |
> > >    | SE               | 0.9758                  | 0.5110           |
> > >    | CE               | 0.9914                  | 0.5730           |
> > >    | JE               | 0.9701                  | 0.6327           |
> > >    | VNE              | 0.9104                  | 0.1022           |
> > >    | Stats (mean$\pm$std) | 0.9675$\pm$0.0263   |                  |
> > >
> > >    Through cross-validation, we observe that even when the target mixture and the fitted mixtures differ significantly (e.g., VNE, which has almost no linear dependency with the others, as indicated by its `near-zero correlation`), the law still demonstrates a high Goodness of Fit, with values over 0.95 and 0.91 for SlimPajama and the Pile, respectively. These results indicate substantial accuracy despite the differing correlations.
> > >
> > > ## Weaknesses Paragraph 2
> > >
> > > 2. Regarding your concerns in **W2** about the insufficient task evaluation in Section 5.3:
> > >
> > >    We made every effort to enhance the evaluation within the limited timeframe and added two new task validations. The following tables present a performance comparison of the 1B models on SQuADv2 and Natural Questions.
> > >
> > >    Table 3: Performance of models trained on the SlimPajama dataset
> > >
> > >    |          | SQuADv2 | Natural Questions |
> > >    |----------|:-------:|:-----------------:|
> > >    | BiMix    | 30.09   | 1.05              |
> > >    | DoReMi   | 29.42   | 0.91              |
> > >    | Baseline | 28.27   | 0.17              |
> > >
> > >    Table 4: Performance of models trained on the Pile dataset
> > >
> > >    |          | SQuADv2 | Natural Questions |
> > >    |----------|:--------:|:-----------------:|
> > >    | BiMix    | 27.15   | 0.90              |
> > >    | DoReMi   | 21.93   | 0.59              |
> > >    | Baseline | 19.08   | 0.55              |
> > >
> > >    It is worth noting that the significant performance disparity between SQuADv2 and Natural Questions is expected, reflecting the varying difficulty of the tasks. For reference, the DoReMi paper reports (in their Table 5) that an 8B model performs 44.43 and 4.35 on these two tasks, respectively.
> > >
> > > ## Weakness Paragraph 3
> > >
> > > 3. The concern raised in **W3** has already been addressed in **Q2**. We will update the manuscript in the next revision cycle to include clarifications for Q1, Q2, Q3.2, and Q5, along with the experimental results presented here.
> > >
> > > ---
> > >
> > > We hope the clarifications adequately address your concerns. We truly appreciate your time and effort in reviewing our work, and we would be extremely grateful if you could consider raising the score to reflect the improvements made. Your positive evaluation would mean a great deal to us.

---

> > > > ### Comment · Reviewer_oXCG · 2024-12-03
> > > >
> > > > Thank you for the response and the additional experiments. However, the new results are still insufficient to fully support the method's reliability.
> > > >
> > > > The leave-one-out cross-validation is a good step, but the mixture proportions remain limited. Also, two tasks are not enough to draw meaningful conclusions (the evaluation is actually straightforward and does not require too many computational resources).
> > > >
> > > > More diverse mixture proportions and evaluations on a broader range of tasks are needed to demonstrate robustness. Please also ensure these results are added to the manuscript for completeness.
> > > >
> > > > That said, I have updated my score to 5 to reflect the improvements made.

---

> > > > > ### Author Response · Authors · 2024-12-04
> > > > >
> > > > > Dear Reviewer oXCG,
> > > > >
> > > > > We genuinely appreciate the time and effort you have dedicated throughout the review process. We will include the clarifications and additional results in the manuscript to enhance its transparency and completeness. Given that the primary contribution of our research lies in the mathematical modeling of data mixing behaviors, we sincerely hope that our findings will offer insightful perspectives to researchers and practitioners, promoting broader interest and deeper exploration in this field.
> > > > >
> > > > > Thank you once again! Your enthusiastic engagement and thoughtful feedback have truly helped us enhance the quality of our manuscript.
> > > > >
> > > > > Sincerely,
> > > > > Authors

---

### Meta-Review · Area_Chair_JUrq · 2024-12-19

**Metareview:**

The paper introduces a novel bivariate data mixing law that models the joint scaling behavior of domain proportions and data volume in LLM pretraining. The authors claim that BiMix provides a systematic framework for understanding and optimizing data mixtures across diverse domains.

Main weaknesses of the paper:
- The empirical examination and application of the proposed data-mixing law are limited. Only two data mixture proportions (baseline and DoReMi) are tested, which may not provide sufficient evidence of general applicability.  While the leave-one-out cross-validation is added in rebuttal, but the mixture proportions remain limited.
- Two downstream tasks are not enough to draw meaningful conclusions. Besides, the evaluation is actually straightforward and does not require too many computational resources.

More diverse mixture proportions and evaluations on a broader range of tasks are needed to demonstrate robustness.

**Additional Comments On Reviewer Discussion:**

- Reviewer oXCG highlighted concerns about the limited empirical examination and application of the proposed data-mixing law. Only two data mixture proportions (baseline and DoReMi) were tested, which may not provide sufficient evidence of general applicability. The reviewer also noted that these two proportions are highly similar, limiting the effectiveness of the verification. The authors added Leave-One-Out Cross-Validation (LOOCV) to offer a reliable and unbiased estimate of model generalization. However, the reviewer thought that the new results are still insufficient to fully support the method's reliability. Furthermore, the reviewer pointed out that two tasks are not enough to draw meaningful conclusions.

- Reviewer VKW2 noted that the paper's downstream evaluation was limited to perplexity (PPL), lacked detailed efficiency comparisons, and did not include comparisons with other recent advancements like RegMix or Online Data Mixing. The authors clarified that the evaluation is not limited to PPL but also includes downstream tasks, detailed in the appendix how both theoretical and experimental evidence support the efficiency of the method. The authors also conducted more experimental comparisons.

---

### Decision · Program_Chairs · 2025-01-22

Reject